# Antioxidant Properties of Four Commonly Consumed Popular Italian Dishes

**DOI:** 10.3390/molecules24081543

**Published:** 2019-04-19

**Authors:** Alessandra Durazzo, Massimo Lucarini, Antonello Santini, Emanuela Camilli, Paolo Gabrielli, Stefania Marconi, Silvia Lisciani, Altero Aguzzi, Loretta Gambelli, Ettore Novellino, Luisa Marletta

**Affiliations:** 1CREA Research Centre for Food and Nutrition, Via Ardeatina 546, 00178 Rome, Italy; emanuela.camilli@crea.gov.it (E.C.); paolo.gabrielli@crea.gov.it (P.G.); stefania.marconi@crea.gov.it (S.M.); silvia.lisciani@crea.gov.it (S.L.); altero.aguzzi@crea.gov.it (A.A.); loretta.gambelli@crea.gov.it (L.G.); luisa.marletta@crea.gov.it (L.M.); 2Department of Pharmacy, University of Napoli Federico II, Via D. Montesano 49, 80131 Napoli, Italy; asantini@unina.it (A.S.); ettore.novellino@unina.it (E.N.)

**Keywords:** Italian popular recipes, food composition database, antioxidant properties, extractable compounds, non-extractable compounds, ferric reducing antioxidant power (FRAP), total polyphenol content (TPC)

## Abstract

Four popular dishes belonging to Italian cuisine and widely consumed in the country were experimentally prepared in a dedicated lab-kitchen following a validated and standardized protocol. This study provides their antioxidant properties evaluating the contribution of extractable and non-extractable bioactive compounds, and identifying the assessment of interactions between their natural active compounds and the food matrix. Ferric reducing antioxidant power (FRAP) values in aqueous-organic extract ranged from the highest antioxidant activity in *torta di mele* (10.72 µmol/g d.m.) to that in *besciamella* (2.47 µmol/g d.m.); in residue, *pasta alla carbonara* reached the highest value (73.83 µmol/g d.m.) following by that in *pasta alla amatriciana* (68.64 µmol/g d.m.). Total polyphenol content (TPC) ranged in aqueous-organic extracts between 36.50 and 64.28 mg/100 g d.m. and in residue from 425.84 to 1747.35 mg/100 g d.m. Our findings may contribute to the updating of the Italian Food Composition Database, by providing for the first time a value for the antioxidant properties. This could contribute to encourage the consumption of recipes rich in key nutrients and bioactive molecules. This information is useful and important for determining the association between diet and a healthy status.

## 1. Introduction

The study of food bioactivity and epidemiological investigations are increasing, emphasizing the perspective of considering the whole food matrix of interest for the risk of disease onset. Following the evolution of nutrition science, currently, researchers are trying to identify the concept of “optimal nutrition” also by studying not only the characteristics and functions of the individual foods or food components, but also their combination in composite foods, dishes, meals, and diets, in order to understand their overall impact on health.

There are only a few foods which are consumed raw, mainly vegetables and fruit, while most foods are heat-treated using different methods chosen according to the matrix, the type of food preparation, and the recipe to prepare them. This aspect is affected by various cultures and culinary traditions. Cooked foods and composed dishes are in fact the most consumed in our daily diet, but there is still little information concerning them, both in terms of their nutritional characteristics and their potential functions, whereas numerous data are available in the literature on the single ingredients, without taking into account either the formulation or the effects of technological process [1]. The interactions between single food components and/or between the different ingredients of a composite dish can play an important role, amplifying the importance of the concept of “food synergy” on health [2]; studies on technological and cooking treatments during the preparation of a recipe also highlight how they can influence their total characteristics, influencing and reflecting the state of health and well-being of consumers. It is well known that the physical, organoleptic, and chemical changes produced in food by heat treatments influence different parameters such as sensorial characteristics, nutrients content, quality and availability, bioactivity, and phytochemical composition [3,4,5,6].

To accurately estimate the dietary intake of the population and prevent cardiovascular disease, cancer, diabetes, etc., in recent years, the focus has increasingly been on studying the nutritional characteristics of foods and traditional recipes that are ready for consumption [7,8,9,10,11,12,13], also with the purpose of formulating dietary recommendations [14,15]. This seems to suggest that there is a need to have comprehensive datasets and databases that include composite and processed foods and food preparations as well as accurate dietary information to investigate the links between diet and health. Currently available data are still limited for two reasons: the scarcity of information on the nutritional composition of commonly consumed foods, in particular processed foods and composite dishes [16], and the deficit of up-to-date knowledge about population dietary habits in different contexts. Food composition databases (FCDBs) are used as the main tool to assess the dietary intake of individuals and groups of people at the regional, national, and international levels [17,18,19,20]. In addition to providing dietary information, consumed food characteristics and their overall nutritional role are also addressed to preserve important cultural elements, such as the great variety of traditional Italian cuisine that distinguishes the gastronomy of the country and reflects both the history and the local characteristics [21].

Specific research projects, such as the European Food Information Resource (EuroFIR) network [22] and the Italian national project Food Quality and Functional (QUALIFU) [23], were created and developed precisely to study, protect, and maintain the significant culture and culinary traditions of a country; Italy, particularly rich with various traditional foods and dishes, has taken part in these projects, and several national traditional recipes widely consumed have been studied, since they play an important role in preserving the local and regional food cultures.

In this framework, the aim of the present work is to provide new information on the antioxidant properties of some Italian recipes with respect to a previous work [12], in terms of extractable and non-extractable compounds, to examine any healthy aspects and identify their potential beneficial role. 

The total antioxidant properties contribute to an assessment of interactions between natural active compounds and other food matrix components of foods, and this can be considered as a first step and as preliminary action towards the comprehension of potential beneficial properties of food matrices from the perspective of healthy food choices [24]. Each food matrix has its own antioxidant capacity that derives from the combined action of carotenoids, lignans, polyphenols, vitamins C and E, etc. Natural antioxidants can show different physiological properties, such as anti-inflammatory, antimicrobial, anti-allergic, anti-atherogenic, anti-thrombotic, cardio-protective, and vasodilatory effects [24,25,26,27,28,29,30,31,32,33]. With respect to antioxidant chemicals, it is worth mentioning that the current review of Yeung et al. [34], based on a scientific literature landscape analysis of works since 1991, concluded that a transition of the scientific interest, shifting from research focused on antioxidant vitamins and minerals to research on antioxidant phytochemicals (plant secondary metabolites), has been observed. In particular, the scientific community recently reached consensus on the distinction between extractable and non-extractable antioxidants: a development and assessment of methodologies was achieved [35]. Antioxidants occur in two forms [35,36]: as easily extractable compounds -free forms that are soluble in aqueous-organic solvents- and as less extractable compounds -bound forms that remain in the residue of aqueous-organic extract-. Their incidence in foodstuffs as raw, cooked, and processed food products was studied [37,38,39,40,41,42,43,44,45,46,47,48]. As remarked previously by Durazzo [35], due to the presence of multiple aspects and factors, it has become difficult to carry out a categorization of the main trends of the contribution of extractable and non-extractable compounds to the total antioxidant properties of major food groups. Generally, it is thought that the analysis of antioxidants in plant foods that remain in residues is necessary. Particular attention in fact should be paid to high fat food matrices [49] and to complex food matrices [12].

The assessment of bioactive compounds to dietary intake is a key issue. Additionally, a proper assessment of the contribution of extractable and non-extractable compounds to the dietary intake is required. Most of the studies available in the literature present daily intakes of extractable polyphenols, whereas little research has been done on the intake of non-extractable polyphenols [50,51,52,53,54], an important fraction contributing to total polyphenol intake [55]. Saura-Calixto et al. [50] estimated the amount of total polyphenols (as extractable and non-extractable polyphenols) consumed in a whole diet (Spanish Mediterranean diet) and their intestinal bioaccessibility: the amount of non-extractable polyphenols was almost double compared to extractable polyphenols. Pérez-Jiménez and Saura-Calixto [51] evaluated non-extractable polyphenols for the 24 most consumed fruit and vegetables in four European countries (France, Germany, The Netherlands, and Spain): macromolecular antioxidants, made up of hydrolysable polyphenols and polymeric proanthocyanidins, are the major contributors (mean value 57%) to the total polyphenol content of fruit and vegetables, and the intake of non-extractable polyphenols was estimated at about 200 mg. In particular, the authors reported that Spain had the highest daily per capita non-extractable polyphenols intake from fruit, whereas the Netherlands had the highest intake derived from vegetable consumption [51]. It is worth mentioning the work of Koehnlein et al. [52] on estimation of the total dietary antioxidant capacity (TDAC) in the Brazilian population: TDAC, evaluated as the ferric-reducing antioxidant power and as the Trolox equivalent antioxidant capacity, was 10.3 and 9.4 mmol/d, respectively. In a further work [53], the same authors compared the phenolic content and the total antioxidant capacity of the 36 most popular Brazilian foods submitted to aqueous extraction or in vitro digestion: after in vitro digestion, cereals, legumes, vegetables, tuberous vegetables, chocolate, and fruits showed higher phenolic contents and higher antioxidant activities than those obtained by aqueous extraction. The digestion caused a reduction in phenolic contents and the antioxidant activities of beverages (red wine, coffee, and yerba mate) [53]. Another work to mention is the research of Faller et al. [54] on the chemical and cellular antioxidant activity of feijoada, a typical Brazilian dish, coupled with an in vitro digestion: bound and residue contributions to total phenolics were 20.9% and 32.2%, respectively, suggesting that phenolics are capable of reaching the colon after the intake.

The overall goal is the development of specified and dedicated databases as well as the inclusion of extractable and non-extractable compounds in current comprehensive and harmonized FCDBs for a better and correct dietary intake assessment. Studies direction [56,57] on this are carried out in eBASIS BioActive Substances in Food Information System [58,59,60]: search protocols and data collection systems are developed to enable the expansion of eBASIS with new quality evaluated data on extractable and non-extractable antioxidants, producing a valuable unique resource [56,57].

The present study evaluates the contribution of extractable and non-extractable bioactive compounds on the antioxidant properties of four popular Italian dishes commonly consumed, previously characterized by their nutrient content [13], to better understand their nutritional role and provide additional dietary information to be included in the next update of the Italian National Food Composition Database (FCDB) of the CREA Research Centre for Food and Nutrition.

## 2. Results and Discussion

In our study, the antioxidant properties of the whole matrix as consumed were studied due to the complexity of the examined matrices. The changes in antioxidants and the interactions between components are correlated to the phytochemical structure and concentration, to the typology of food matrixes, to the preparation procedure, and to the typology and time of cooking [61,62,63,64]. 

In Table 1, Ferric Reducing Antioxidant Power (FRAP) values (µmol/g d.m.) and Total Polyphenol Content (TPC) (mg/100 g d.m.) were reported for selected popular Italian dishes.

FRAP values in aqueous-organic extract decreased in the following order: *torta di mele* > *pasta alla amatriciana* > *pasta alla carbonara* = *besciamella*; in residue, *pasta alla carbonara* reached the highest value followed by *pasta alla amatriciana*. The FRAP values of pasta-based dishes confirmed the value reported in our previous work [12] for another dish, the *spaghetti alle vongole*, namely 4.20 µmol/g d.m. and 64.22 µmol/g d.m. in aqueous-organic extract and residue, respectively. In general, the antioxidant properties of cereals and derivatives thereof have been well documented over the years [42,65,66]; with reference to antioxidants in pasta, one of most popular staple foods, the effect of cooking [67,68] as well as new formulations and functional products have been studied [69,70,71,72]. In this regard, it is worth mentioning the work of Ioannou et al. [63], since they remarked how the addition of ingredients with high antioxidant activity to a complex preparation can contribute to increases in total antioxidant capacity, but not in a proportional way.

*Besciamella* belongs to the category of the white sauces, and it is an example of milk-based dishes. In recent years, emerging studies on the antioxidant properties of milk [73,74] as well as dairy products [75,76] have been carried out. For instance, Manzi and Durazzo [74], by evaluating the antioxidant properties of industrial heat-treated milk, namely UHT, microfiltered, and high quality pasteurized milk, showed that UHT milk has the highest total polyphenol content, DPPH, and FRAP values. The authors [74] explained that the behavior of UHT milk is probably related to the development of antioxidant compounds, formed during the Maillard reaction occurring when milk treatment is performed at high temperatures (≥135 °C for at least 1 s) according to previous authors [77,78,79].

*Torta di mele* belongs to the subcategory of fruit cake comprised among the desserts, and evidenced the highest FRAP value in aqueous-organic extract and the lowest value in the residue with respect to other items. This seems to reflect the high content of organic acids in the apples used in the preparation of this dessert [80,81]. For instance, for apple, in Phenol Explorer Databases the total polyphenol content was 131.80 mg/100 g FW, as mean of 53 original content values extracted from eight published papers [82].

Table 2 shows and summarizes the contribution of extractable and non-extractable compounds to the antioxidant properties of all Italian dishes we investigated grouped by category, in this work and in a previous one [12].

For the four new dishes studied, the extractable antioxidants (aqueous-organic extracts) were minor contributors to the total antioxidant activity; consequently, the hydrolysable polyphenols (residues) contributed significantly more: for *pasta alla amatriciana*, *pasta alla carbonara*, and *besciamella* the extractable antioxidants contribute less than 6% and non-extractable compounds contribute in a range from 94 to 97%, whereas for *torta di mele* the percentages of contributions were 37% and 63%, respectively, for extractable polyphenols and hydrolysable polyphenols.

Regarding total polyphenol content evaluation, as reported in Table 1, TPC values ranged in aqueous-organic extracts between 36.50 and 64.28 mg/100 g d.m. and in residue from 425.84 to 1747.35 mg/100 g d.m. Hydrolysable polyphenols represent consequently a significant fraction, by accounting in the range between 87% (*torta di mele*) and 98% (*pasta alla carbonara*) of total polyphenols. A good Pearson correlation between TPC values with FRAP ones was found in aqueous-organic extract (r = 0.6798) and in residues (r = 0.9909). For instance, *pasta alla carbonara* showed the lowest values in aqueous-organic extract and the highest value in residue both in FRAP and TPC, whereas *torta di mele* showed the opposite behavior.

## 3. Materials and Methods

### 3.1. Recipes: Identification of Standard Recipe, Sampling and Dish Preparation

Four recipes (Table 3), all identified from survey Italian National Food Consumption Survey INRAN-SCAI 2005-2006 [83], and selected in the QUALIFU project [23], were experimentally prepared [13] in a dedicated lab-kitchen following a validated and standardized protocol developed within the EuroFIR Network [84]; three preparations represented some popular, most commonly consumed Italian dishes: *pasta alla amatriciana*; *pasta alla carbonara*, and *torta di mele*; one sauce, the *besciamelle*, was selected and studied for its great use in other traditional food preparations (lasagna, vegetables au gratin, baked pasta, salted cakes, etc.). The different ingredients included were pasta, cured meat, milk, cheeses, eggs, vegetables, fruits, extra virgin olive oil, and butter, thus covering a wide range of antioxidant properties. Our previous studies on traditional Italian dishes were focused on determining proximate composition and a dietary intake evaluation [12,13] and aimed at applying an integrated and emerging (analytical) approach to classifying dishes [85,86]. The present study focuses on the health-beneficial properties of selected traditional Italian dishes with the aim of extending and triggering interest to this type of research, which connects the nutritional aspects to health-beneficial properties and traditionally consumed foods.

In detail, at first a document collection was carried out from the most popular and traditional cookbooks in Italy (Il cucchiaio d’argento; La cucina italiana, etc.) for every recipe selected; therefore, for every dish one “standard recipe” was identified and one “preparation protocol” was drafted in detail identifying ingredients, quantities, preparation techniques, type, temperature, and time cooking; dishes production had been carried out according to the standard procedures developed within the EuroFIR network.

The sampling plan had taken into account the collection of simple ingredients at various retail stores and supermarkets in Rome; to represent the variability of the ingredients, the main brands and cultivars of the same product were considered and purchased: 8 brands of pasta and tomato pulp, 7 brands of eggs, 6 brands of bacon, 4 samples of Amatrice cheek lard and Pecorino cheese, 4 brands of extra virgin olive oil, and 2 brands of wheat flour, butter, and milk. Each brand (primary sample) for every ingredient (secondary sample) was properly prepared, weighed, and then combined to make a composite sample (pool) before use for the preparation of the final dish (laboratory sample). This was weighed, assembled, and cooked in a laboratory and dedicated kitchen at the CREA Research Centre for Food and Nutrition by trained persons according to the preparation protocol of the standard recipe, applying methods and utensils commonly used in households. The recipes, once completed and cooked, were homogenized, frozen at −30 °C, and lyophilized. For each type of dish, two independent batches (laboratory sample) were prepared and on each one the analysis were performed in triplicate.

### 3.2. Evaluation of Antioxidant Properties by Ferric Reducing Antioxidant Power (FRAP) and Total Polyphenol Content (TPC)

#### 3.2.1. Extraction Procedure

Extractable and non-extractable polyphenols were extracted as described by Durazzo et al. [12]. Aqueous-organic extracts (extractable antioxidants) and their residues (non-extractable antioxidants) were isolated and studied.

In particular, among non-extractable antioxidants, attention was paid to hydrolysable polyphenols, which were isolated and determined following a specific and suitable acid hydrolysis procedure as reported below.

##### Aqueous-Organic Extract

On the basis of dish ingredients and available literature data, a quantity of 3–5.5 g of each sample was placed in a test tube, and 20 mL of acid methanol/water (50:50 *v/v*, pH 2) were added. The tubes were vortexed at room temperature for 3 min and then mildly shaken for 1 h in a water bath at room temperature. The tubes were then centrifuged at 2500 rpm for 10 min and the supernatants were recovered. Twenty milliliters of acetone/water (70:30 *v/v*) mixture were added to the residues. All operations (vortexing, shaking, centrifugation) were then repeated. Methanolic and acetonic extracts were combined and centrifuged at 2800 rpm for 15 min. The resulting supernatant was transferred into tubes and directly used for the determination of FRAP and TPC.

##### Residue

The residues remaining after the previously described extraction were left in a ventilated and heated apparatus (max temperature 25 °C) until dry. Briefly, about 200–450 mg of the residue, respectively, were mixed with 20 mL of methanol and 2 mL of concentrated sulfuric acid (18 M). The samples were gently stirred for 1 min and were shaken in a water bath at 85 °C for 20 h; samples were then centrifuged (2500 g for 10 min), and the supernatant was recovered. After two washings, with minimum volumes of distilled water and recentrifuging where necessary, the final volume was 50 mL. The tube was centrifuged at 2800 rpm for 20 min, and the resultant supernatant was directly used for the determination of FRAP and TPC.

#### 3.2.2. Antioxidant Assays

Several methods have been proposed for evaluating the antioxidant properties of single compounds and foods [87]. The most common are (i) the Folin–Ciocalteu assay used widely to determine the total phenolics; (ii) the Trolox equivalent antioxidant capacity (TEAC); (iii) the oxygen radical absorbance capacity (ORAC); (iv) the total radical-trapping antioxidant parameter (TRAP); (v) the ferric-reducing antioxidant power (FRAP); and (vi) the 2,2-diphenyl-1-picrylhydrazyl (DPPH) radical scavenging activity assay [88,89,90]. All these methods are based on the measurement of the capacity of a food component or a food to scavenge specific free radicals or to reduce a target molecule.

These assays differ in their principles, mechanisms, and experimental conditions as well as in how their end points are measured, so different methods to estimate and/or determine the antioxidant activity of the compounds should be carried out. Three are the main mechanisms by which the antioxidants act, encompassing the direct reaction with radicals and the chelation of free metals (involved in reaction finally generating free radicals): the H atom transfer, the single electron transfer, and the metal chelation [91]. Literature data report that the use of at least two or three assays is strongly recommended for assessing antioxidant properties [92]. Prior et al. [93] proposed that procedures and applications for three assays should be considered for standardization: the oxygen radical absorbance capacity (ORAC) assay, the Folin–Ciocalteu method, and possibly the Trolox equivalent antioxidant capacity (TEAC) assay [93].

##### FRAP

The determination of the FRAP assay was carried out according to the methods of Benzie & Strain [94] and Pulido et al. [95], through the use of a Tecan Sunrise^®^ plate reader spectrophotometer. The method is based on the reduction of the Fe^3+-^TPTZ (2,4,6-tripyridyl-s-triazine) complex to a ferrous one at acidic pH value. 

##### Total Polyphenol Content (TPC)

The TPC was determined using the Folin–Ciocalteu method [96]. Briefly, appropriate dilutions of extracts were oxidized with Folin–Ciocalteu reagent, and the reaction was neutralized with sodium carbonate. The absorbance of the resulting blue color was measured at a wavelength of 760 nm against an appropriate blank after 2 h of reaction at room temperature. Gallic acid was used as reference standard.

### 3.3. Statistical Analysis

All analyses were performed in triplicate. Data are presented as mean ± standard deviation (s.d.) of the analysis carried out on two preparations of every dish. Statistica for Windows (Statistical package; release 4.5; StatSoft Inc., Vigonza, PD, Italy) was used to perform one-way analysis of variance (ANOVA) and a post-hoc test: Tukey’s honest significant difference (HSD) test.

## 4. Conclusions

In this study, antioxidant properties of four commonly consumed popular Italian dishes are provided for the first time. Our findings can contribute to the updating of the Italian FCDB by providing a value of antioxidant properties that are useful and important for study on the association between diet and a healthy status [97,98].

The innovative character of this research lays in the fact that the four dishes were experimentally prepared in a dedicated lab-kitchen following a validated and standardized protocol based on harmonized guidelines. The other key aspect is the study of antioxidant properties, in term of extractable and non-extractable antioxidants, applied to complex matrixes, i.e., food preparations and food composite dishes.

Our study highlighted the importance of evaluating the real nutritional information about foods as taken, since ingredients are often mixed and heat-treated to formulate/prepare dishes. The availability of these new and appropriate food composition data is needed in order to correctly evaluate the dietary intake of recipes rich in key nutrients and bioactive molecules, facilitating further nutrition-related studies, and can be used to encourage the consumption of certain recipes. Further studies in this direction are needed to provide a detailed nutritional overview of popular and traditional Italian dishes and are currently being carried out in our laboratories.

## Figures and Tables

**Table 1 molecules-24-01543-t001:** Ferric Reducing Antioxidant Power (FRAP) and Total Polyphenol Content (TPC) of popular Italian dishes *.

	FRAP (µmol/g d.m.)		TPC (mg/100 g d.m.)	
	Aqueous-Organic Extract	Residue	Aqueous-Organic Extract	Residue
*Pasta alla amatriciana*	4.01 ± 0.67 ^b^	68.64 ± 4.43 ^c^	60.87 ± 5.48 ^c^	1447.59 ± 70.33 ^c^
*Pasta alla carbonara*	2.62 ± 0.53 ^a^	73.83 ± 3.52 ^d^	36.50 ± 6.31 ^a^	1747.35 ± 72.91 ^d^
*Besciamella*	2.47 ± 0.17 ^a^	52.98 ± 1.22 ^b^	51.90 ± 3.38 ^b^	1173.44 ± 73.07 ^b^
*Torta di mele*	10.72 ± 0.80 ^c^	18.24 ± 5.09 ^a^	64.28 ± 2.39 ^c^	425.84 ± 63.86 ^a^

* Mean ± S.D.; ANOVA and Tukey’s HSD test: by column, means followed by different letters are significantly different (*p* < 0.05).

**Table 2 molecules-24-01543-t002:** Contribution % of extractable and non-extractable compounds to antioxidant properties of popular Italian dishes *.

Italian Dishes	Aqueous-Organic Extract	Residue
Sauces		
*Besciamella*	4	96
First Courses		
*Spaghetti alle vongole **	6	94
*Pasta alla amatriciana*	6	94
*Pasta alla carbonara*	3	97
One Dish Meals		
*Pomodori al riso **	15	85
*Gâteau di patate **	11	89
Side Courses		
*Carciofi alla romana **	58	42
Desserts		
*Pan di Spagna **	5	95
*Torta di mele*	37	63

* Data derived from Durazzo et al. [12].

**Table 3 molecules-24-01543-t003:** Italian popular dishes: ingredients, method and time cooking.

Original Name	Food Name	Ingredients (g/100 g)	Cooking	Timing (min.)
*Pasta alla amatriciana*	Amatriciana pasta	Short pasta (37.5), tomato pulp (37.5), Amatrice cheek lard diced (16), Amatrice Pecorino cheese PAT (hard cheese from sheep) (7.5), extra virgin olive oil (1.1), salt (0.3), chili pepper (0.1).	Boiling, pan-frying, and simmering	25
*Pasta alla carbonara*	Carbonara pasta	Short pasta (47.3), bacon cubes (20.3), Roman Pecorino cheese PDO (hard cheese from sheep) (13.6), eggs (16.6), extra virgin olive oil (1.4), salt (0.4), black pepper (0.4).	Boiling and pan-frying	13
*Besciamella*	Béchamel sauce	Milk (83), butter (8), flour (8), salt (0.5).	Simmering gently	33
*Torta di mele*	Apple Pie	Apples (37.7), sugar (15), wheat flour (18), butter (9.3), eggs (9.2), whole milk (7.5), baking powder (1), vanilla (0.03), grated lemon peel (0.2), lemon juice (2.4).	Baking	30

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
