# Peer review of "Antioxidant Properties of Four Commonly Consumed Popular Italian Dishes"

_molecules, 2019, doi:10.3390/molecules24081543_

Round 1

Reviewer 1 Report

Authors have properly addressed all my questions and clearly improved the quality of this short communication.

They only have to check and revise English for grammar and some typo errors in the new parts included during the revision (lines 113-139; lines 234-252).

Author Response

Inizio modulo

The authors want to express their gratitude to the reviewer for his/her valuable comments and suggestions. The authors’ replies to the individual points raised are reported in Italic below.

pen Review

English language and style

( ) Extensive editing of English language and style required 
(x) Moderate English changes required 
( ) English language and style are fine/minor spell check required 
( ) I don't feel qualified to judge about the English language and style 

Yes

Can be improved

Must be improved

Not applicable

Does the   introduction provide sufficient background and include all relevant   references?

(x)

( )

( )

( )

Is the   research design appropriate?

(x)

( )

( )

( )

Are the   methods adequately described?

(x)

( )

( )

( )

Are the   results clearly presented?

(x)

( )

( )

( )

Are the   conclusions supported by the results?

(x)

( )

( )

( )

Comments and Suggestions for Authors

Authors have properly addressed all my questions and clearly improved the quality of this short communication.

They only have to check and revise English for grammar and some typo errors in the new parts included during the revision (lines 113-139; lines 234-252).

The linguistic revision of manuscript was carried out including the specified parts.

Submission Date

11 March 2019

Date of this review

12 Mar 2019 18:40:36

Reviewer 2 Report

The present manuscript described the total phenolic content and antioxidant activity of common italian dishes. Although the topic is interest, the implementation is quite weakly.

The number of dishes is too short (n=4)

They used only simple, spectrometric assays that give a rapid estimation.

the parameters studied cannot be linked with specific health claims. EFSA and USFDA reject total phenolic content and antioxidant activity as markers of bioactivity. Thus, these measurements are only prelimenary data.

Aurhors reported that a synergistic effect may occurs in dishes. Thus, the phytochemical composition and antioxidant activity of ingredients have to be assessed and to investigate the synergistic or antagonistic effect.

Minor comments:

 the introduction is too long

the paragraph 2.2.2 can be removed, the chemitry behind these assays is well-known.

Author Response

The authors want to express their gratitude to the reviewer for his/her valuable comments and suggestions. The authors’ replies to the individual points raised are reported in Italic below.

Open Review

English language and style

( ) Extensive editing of English language and style required 
( ) Moderate English changes required 
( ) English language and style are fine/minor spell check required 
(x) I don't feel qualified to judge about the English language and style 

Yes

Can be improved

Must be improved

Not applicable

Does the   introduction provide sufficient background and include all relevant   references?

(x)

( )

( )

( )

Is the research   design appropriate?

( )

( )

(x)

( )

Are the methods   adequately described?

(x)

( )

( )

( )

Are the results   clearly presented?

(x)

( )

( )

( )

Are the   conclusions supported by the results?

(x)

( )

( )

( )

Comments and Suggestions for Authors

The present manuscript described the total phenolic content and antioxidant activity of common italian dishes. Although the topic is interest, the implementation is quite weakly.

The number of dishes is too short (n=4)

As marked in the text this work is a short communication that wants “to provide a new update of the antioxidant properties of some Italian recipes respect to a previously work [12], in terms of extractable and non-extractable compounds, to examine any healthy aspects and identify their potential beneficial role”. Moreover the samples are food preparation that were experimentally prepared in a dedicated lab-kitchen following a validated and standardized protocol based on harmonized guidelines: this represent a key experimental issue of research.

They used only simple, spectrometric assays that give a rapid estimation.

As marked in the conclusion the “key aspect is the study of antioxidant properties, in term of extractable and non-extractable antioxidant, applied to complex matrixes, i.e. food preparations and food composite dishes”.

the parameters studied cannot be linked with specific health claims. EFSA and USFDA reject total phenolic content and antioxidant activity as markers of bioactivity. Thus, these measurements are only prelimenary data.

As  marked in work  “The total antioxidant properties identify the assessment of interactions between natural active compounds and other food matrix components of foods and they can be considered as the first step and preliminary action for the comprehension of potential beneficial properties of food matrices in the perspective of healthy choices”.

Aurhors reported that a synergistic effect may occurs in dishes. Thus, the phytochemical composition and antioxidant activity of ingredients have to be assessed and to investigate the synergistic or antagonistic effect.

As marked in the work “numerous data are available in literature on the single ingredients, without taking into account either the formulation or the effects of technological process [1]. The interactions between single food components and/or between the different ingredients of a composite dish can play an important role, amplifying the importance of the concept of "food synergy" on health [2]; also studies on technological and cooking treatments during the preparation of a recipe highlight how they can influence their total characteristics, reflecting on the state of health and well-being of people. It is well known that the physical, organoleptic and chemical changes produced in food by heat treatments, influence different parameters such as sensorial characteristic, nutrient content, quality and availability, bioactivity and phytochemical composition [3-5]”. Indeed multi-factors should be considered: the ingredients, the preparation procedure, each steps of cooking –type and time-, in this order a proper approach of analysis is to evaluate the antioxidant properties on food preparations as consumed. As previous marked “Most people consume a combination of different foods containing a large variety of nutrients and bioactive components. This calls for a holistic approach because identifying healthy nutritional combinations including mixed dishes is more appropriate than analyzing the effects of isolated individual ingredients. Few studies have been carried out on the antioxidant properties of complex food matrices and ready-to-eat dishes, whereas numerous data on single ingredients are available in literature, but unfortunately they do not take into account either the formulation or the effects of the technological process”.

Minor comments:

 the introduction is too long

the paragraph 2.2.2 can be removed, the chemitry behind these assays is well-known.

Both the insertion of additional part in introduction and the paragraph 2.2.2 was requested from other Reviewer

Reviewer 3 Report

This article is important from the gastronomic point of view, as it gives a nutritional value in the form of antioxidants in four popular Italian dishes. I also received an article with comments, that I agree with. In addition to the above, I would suggest clarifying the methodology of work, specifically to define whether analyzes were made from only one portion of the same meal (3 reps), or more portions of the same meal were prepared and analyzes were performed three times.

Author Response

The authors want to express their gratitude to the reviewer for his/her valuable comments and suggestions. The authors’ replies to the individual points raised are reported in Italic below.

Review Report Form

Open Review

English language and style

( ) Extensive editing of English language and style required 
( ) Moderate English changes required 
( ) English language and style are fine/minor spell check required 
(x) I don't feel qualified to judge about the English language and style 

Yes

Can be improved

Must be improved

Not applicable

Does the   introduction provide sufficient background and include all relevant   references?

(x)

( )

( )

( )

Is the research design   appropriate?

(x)

( )

( )

( )

Are the methods   adequately described?

( )

(x)

( )

( )

Are the results   clearly presented?

(x)

( )

( )

( )

Are the   conclusions supported by the results?

(x)

( )

( )

( )

Comments and Suggestions for Authors

This article is important from the gastronomic point of view, as it gives a nutritional value in the form of antioxidants in four popular Italian dishes. I also received an article with comments, that I agree with. In addition to the above, I would suggest clarifying the methodology of work, specifically to define whether analyzes were made from only one portion of the same meal (3 reps), or more portions of the same meal were prepared and analyzes were performed three times.

As you suggested, it was better clarified in the text: For each type of dish, two independent batches (laboratory sample) were prepared and on each one, the analysis were performed three times.

Submission Date

11 March 2019

Date of this review

18 Mar 2019 12:44:00

Reviewer 4 Report

This manuscript provided the antioxidant properties evaluating four commonly consumed popular Italian dishes. The methods and results were well shown, and results contribute to the updating of Italian Food Composition Database. I suggest accept it after minor revisions.

The present title is not meet the key contents of manuscript. I suggest the title should be revised as “Antioxidant properties of four commonly consumed popular Italian dishes”.

Author Response

The authors want to express their gratitude to the reviewer for his/her valuable comments and suggestions. The authors’ replies to the individual points raised are reported in Italic below.

Review Report Form

Open Review

English language and style

( ) Extensive editing of English language and style required 
( ) Moderate English changes required 
(x) English language and style are fine/minor spell check required 
( ) I don't feel qualified to judge about the English language and style 

Yes

Can be improved

Must be improved

Not applicable

Does the   introduction provide sufficient background and include all relevant   references?

(x)

( )

( )

( )

Is the research design   appropriate?

( )

(x)

( )

( )

Are the methods   adequately described?

(x)

( )

( )

( )

Are the results   clearly presented?

( )

(x)

( )

( )

Are the   conclusions supported by the results?

( )

(x)

( )

( )

Comments and Suggestions for Authors

This manuscript provided the antioxidant properties evaluating four commonly consumed popular Italian dishes. The methods and results were well shown, and results contribute to the updating of Italian Food Composition Database. I suggest accept it after minor revisions.

The present title is not meet the key contents of manuscript. I suggest the title should be revised as “Antioxidant properties of four commonly consumed popular Italian dishes”.

As you suggested, the title was changed into “Antioxidant properties of four commonly consumed popular Italian dishes”

Reviewer 5 Report

Paper needs some editing in order to correct for mistakes through the text. All of my comments are posted through text, so please see attached file.

I have to say that this paper can be accepted after minor revisions.

Author Response

The authors want to express their gratitude to the reviewer for his/her valuable comments and suggestions. The authors’ replies to the individual points raised are reported in Italic below.

Review Report Form

Open Review

English language and style

( ) Extensive editing of English language and style required 
( ) Moderate English changes required 
(x) English language and style are fine/minor spell check required 
( ) I don't feel qualified to judge about the English language and style 

Yes

Can be improved

Must be improved

Not applicable

Does the   introduction provide sufficient background and include all relevant   references?

(x)

( )

( )

( )

Is the research design   appropriate?

(x)

( )

( )

( )

Are the methods   adequately described?

(x)

( )

( )

( )

Are the results   clearly presented?

(x)

( )

( )

( )

Are the   conclusions supported by the results?

(x)

( )

( )

( )

Comments and Suggestions for Authors

Paper needs some editing in order to correct for mistakes through the text. All of my comments are posted through text, so please see attached file.

I have to say that this paper can be accepted after minor revisions.

All the comments suggested in pdf file have been followed and were inserted in the text.

Round 2

Reviewer 2 Report

x       

This manuscript is a resubmission of an earlier submission. The following is a list of the peer review reports and author responses from that submission.

Round 1

Reviewer 1 Report

This is an interesting manuscript on the antioxidant capacity and total polyphenols content in four traditional Italian dishes intended to filling the gap of scarcity of these kind of data in FCDB. The significance of content and interest for readers is very high since can provide data of interesnt for choosing high antioxidant content recipes and facilitate nutrition and intervention related studies. However it needs to be improved addressing some concerns before to be published.

1. Line 56. A recent relevant review on food processing and bioaccessibility of bioactive compounds should be incorporated to this text. (Cilla et al. 2018 J Food Compos Anal 68, 3-15).

2. Lines 70-71 aprox. Check the number of these references and subsequent ones since do not match with the list of references (20-23).

3. In the introduction some references related to this topic are missing and should be included and discussed.

-Koehnlein et al. 2014 Int J Food Sci Nutr 65, 293-298

-Koehnlein et al. 2016 Int J Food Sci Nutr 67, 614-623

-Kremer Faller et al. 2012 J Agric Food Chem 60, 4826-4832

-Saura-Calixto et al. 2007 Food Chem 101, 492-501

4. Line 179. Why FRAP and only the use of one total antioxidant capacity method was selected? According to a seminal review (Prior et al. 2005 J Agric Food Chem 53, 4290-4302) on total antioxidant capacity methods ORAC, TPC and TEAC should be of election. Discuss in text.

5. Line 185. Try to compare data on TPC using Phenol-Explorer database if possible. Indicate in table 2 that TPC are expressed as mg GAE/100g d.m.

6. Line 195. Indicate the post-hoc test after ANOVA

7. Lines 221-222. Maybe these compounds can be Maillard reaction compounds? Clarify and discuss.

8. Line 228. Reference should be quoted with number.

9. Conclusions. Some studies relating dietary total antioxidant capacity and helath are required to be included to give strenght to the results here obtained.

10. Line 303. Incomplete bibliographic data. Include at leat the doi number if the article is in press.

Reviewer 2 Report

The purpose of the work was to analyze the antioxidant capacity and total phenolic compounds of four Italian dishes. I leave it to the editor's decision, but I suggest that the article be rejected. The analytical methods used in the work are very standard. Molecules publish research using analytical techniques that enable the identification of individual compounds contained in food. Determining the antioxidant capacity of four traditional dishes does not provide significant information for food science. The article would be interesting, as in addition to the antioxidant capacity and the total polyphenol content, the content of other compounds affecting antioxidant properties (eg vitamin C, carotenoids, tocopherols, sterols) was determined. Also, the work would be more interesting if the determination of polyphenols was performed by chromatographic method.